# Bottleneck and enabler evaluation of avian influenza health event — Guatemala, January-February 2023

Parsa Bastani[1], Edgar Bailey Leonardo[2], Jose Carlos Monzon Fuentes[2], Cesár Conde Pereira[3], Emily Zielinski Gutierrez[2], Parminder S. Suchdev[2]*

1 Epidemic Intelligence Service, Centers for Disease Control and Prevention, Atlanta, Georgia, United States of America, 2 Central America Office, Division of Global Health Protection, Centers for Disease Control and Prevention, Guatemala City, Guatemala, 3 Ministry of Public Health and Social Assistance, Guatemala

* psuchdev@cdc.gov

## Abstract

In February 2023, H5N1 was identified in 11 wild pelicans in Izabal, Guatemala. These were the first known cases of H5N1 in the country. This study assessed the timeliness of the response to this One Health event using the "7-1-7" benchmarks, which propose the following metrics: detection within seven days, notification within one day, and completion of early response within seven days. Open-ended interviews were conducted in September 2023 with nine key informants from the Ministry of Agriculture, Livestock, and Food (MAGA) and the Ministry of Health and Social Assistance (MSPAS) who were directly involved in the response. Participants included epidemiologists, laboratory analysts, and other relevant personnel. Interviews were analyzed using UNICEF's "Human-Centered Design 4 Health" approach to qualitative fieldwork. Detection and notification were completed in one day, while early response was completed in 34 days. Key enablers of the response included interregional notification and cooperation, availability of earmarked emergency funds, event-based surveillance, and support from laboratories across public and private sectors. Reported bottlenecks included limited national laboratory testing capacity, challenges in inter-agency and intra-agency communication, workforce constraints, and equipment shortages. This outbreak response met the detection and notification criteria but did not achieve the 7-day target for completing early response activities. As one of the few qualitative studies examining avian influenza response in Central America, these findings highlight how strengthening a One Health approach, particularly in communication, workforce, and laboratory capacity, could enhance preparedness for future outbreaks.

**Data availability statement:** All relevant data are within the paper.

**Funding:** This work was funded by the Centers for Disease Control & Prevention (PS, EBL, JCMF, EZG, PSS). The funders had no role in study design, data collection and analysis, decision to publish, or preparation of the manuscript. Additional disclosure: The findings and conclusions in this report are those of the authors and do not necessarily represent the official position of the US Centers for Disease Control and Prevention.

**Competing interests:** The authors have declared that no competing interests exist.

## Introduction

Avian influenza outbreaks are a persistent global health concern, affecting both animal and human populations. Specifically, highly pathogenic avian influenza (HPAI) is a highly contagious disease that can be fatal in birds [1]. Humans with close or prolonged, unprotected exposures are at a greater risk of infection. Human illnesses range from mild to severe, sometimes leading to death [2]. In 2022 and 2023, outbreaks of H5N1, a strain of HPAI, were identified in both domestic and wild birds as the virus spread among migratory waterfowl from Canada and the United States to Mexico, Central, and South America [3]. The virus spread mainly along the Pacific flyway and resulted in the first occurrence of HPAI in Argentina, Bolivia, Colombia, Costa Rica, Cuba, Ecuador, Guatemala, Honduras, Panama, Peru, Uruguay, and Venezuela [4]. The increased number of outbreaks in bird populations across this region raised concerns that the virus is becoming more fit and adept at spreading [5].

In January 2023, an outbreak of H5N1was suspected in 11 brown pelicans (*Pelecanus occidentalis*) in Izabal, Guatemala, a coastal department that abuts the Caribbean Sea. Local fishermen noticed these sick pelicans and alerted a regional epidemiologist stationed in Izabal. In collaboration with partners, H5N1 was confirmed, and the Guatemalan government coordinated several actions to contain and manage the outbreak. While there was no human spillover, these cases of H5N1 in Guatemala underscore the need to strengthen health security measures and ensure adequate public health systems are in place to address potential outbreaks.

Specifically, early disease detection and response are necessary to contain HPAI outbreaks and minimize their potential spread to humans. One tool available in helping countries identify gaps in their preparedness systems and strategies is the 7-1-7 metrics [6,7]. This is a set of indicators that measure the timeliness of detection, notification, and early response activities. The goal of the 7-1-7 framework is to ensure governments can meet three key timeliness metrics during an outbreak: detect within 7 days of emergence, report to a public health authority responsible for action within 1 day of detection, and complete early response actions within 7 days from reporting to public health authorities. These targets were derived from analyses of past outbreak timeliness data and are aligned with the World Health Organization's Integrated Disease Surveillance and Response (IDSR) Technical Guidelines (3rd Edition) [7,8]. This approach helps rectify the shortcomings of existing pandemic preparedness measures, which often fail to capture how emergency response systems function in real-time. For example, scores on the WHO Joint External Evaluation (JEE) and Global Health Security Index were not correlated with COVID-19 outcomes, such as standardized infection rates [9]. To address this gap, in August 2023, the WHO incorporated 7-1-7 targets into its early action review (EAR) guidance to help countries evaluate the timeliness of their early detection and response activities [10].

A qualitative understanding of issues that serve as bottlenecks and enablers for timely outbreak response is key to achieving 7-1-7 targets. We define these targets as the following: detection was when local residents detected the aberrant health event of sick pelicans along the coast; notification was when the event was reported

to the Guatemalan Ministry of Agriculture, Livestock and Food; and early response completion was when the investigation team was deployed, initial risk assessments were completed, laboratory confirmation of the outbreak etiology was obtained, and public health measures, risk communication, and community engagement measures were initiated by human and animal health sectors. This information can help governments direct where to channel funding and resources for improving their outbreak preparedness. Thus far, one retrospective study has been published on identifying bottlenecks and enablers to 7-1-7 system performance [6]. In this study, data collection primarily took place through reviewing situations, intra-action and after-action reviews, discussions with subnational and national-level responders, and a 1-day workshop. Given the limited research on 7-1-7 bottlenecks and enablers, additional tools are needed for evaluating outbreak situations with limited written records that use the 7-1-7 framework.

This study expands upon extant 7-1-7 literature by focusing on a country in Central America, examining an animal outbreak of a zoonotic disease prior to a potential spill-over event, and proposing a unique, simple, and rapid qualitative methodology for collecting and analyzing data on health emergency preparedness. Specifically, this study systematizes the collection of qualitative materials through semi-structured interviews with key officials. Our aims were twofold: 1) to calculate 7-1-7 timeliness metrics during the January H5N1 outbreak in Guatemala, and 2) to identify bottlenecks and enablers that affected this outbreak response.

## Methods

### Ethics statement

This activity was reviewed by the US Centers for Disease Control and Prevention (CDC) and was deemed to be non-human subjects research. Verbal consent was obtained for all participants.

### Study design and data collection

Recruitment and interviews were conducted between 17/09/2023 and 30/09/2023 with nine key informants in Guatemala involved in the H5N1 response using a purposive sampling strategy. Key informants were selected from government agencies that were most involved in the outbreak. These included epidemiologists, laboratory analysts, and program officers from the Guatemalan Ministry of Agriculture, Livestock and Food (MAGA) and the Ministry of Public Health and Social Assistance (MSPAS). Due to high staff turnover, it was not possible to locate and consent further respondents within the recruitment time frame. Interviews were conducted in Spanish and English and lasted between 30–60 minutes. Interviews were conducted either on the phone or in-person in a private location in an office space by PB (an epidemiologist and medical anthropologist) and ELB (an epidemiologist and veterinarian). All respondents were verbally consented prior to each interview and consent was documented in the written field notes. Semi-structured interview guides were administered to each respondent (S1 Text). Questions covered the research participant's experience and perceptions of the outbreak response during the detection, notification, and response stages. Each interview was recorded and transcribed.

### Data analysis

Analysis of interview data was conducted using the rapid inquiry approach published in United Nations International Children's Emergency Fund's (UNICEF) "Human-Centered Design 4 Health." [11] This toolkit was originally designed to reduce the technical barriers for conducting qualitative research on basic health services and has been used in several published studies [12–15]. In other words, UNICEF designed this toolkit so that people with no technical training could perform qualitative research in their communities.

We drew on the toolkit's analyses framework using its "Record Field Research" and "Synthesis Tools" worksheets to identify important themes from the transcripts. We followed UNICEF's overall analysis process: first, we filled out one "Record Field Research" worksheet per interview to analyze each of the interviews and second, we filled out one

"Synthesis Tool" per interview subgroup. The first step comprised of listing the following for each interview: five meaningful quotes, five surprising moments, five suggestions from the interviewer, and five interventions the analyzer (PB and EBL) identified based on the interview (S1 Table). Second, we placed 'Record Field Research' worksheets into subgroups based on the relevant backgrounds of respondents. Accordingly, we created four subgroups: 1) national level MAGA staff; 2) local level MAGA staff; 3) national level MSPAS staff, and 4) local level MSPAS staff. Last, we completed one "Synthesis Tool" worksheet for each subgroup by reviewing the 'Record Field Research' worksheets for that subgroup. This comprised of listing the following per subgroup: three powerful quotes, three things that worked well, three suggestions, three specific issues that need to be addressed, and the subgroup's perception of the overall detection, notification, and response to the outbreak (S1 Table). From the five meaningful quotes identified per interview, the three most representative and compelling quotes were selected during the subgroup-level synthesis stage, following the UNICEF toolkit framework but extended in our study through systematic comparison across subgroups. In the UNICEF methodology, the completion of this "Synthesis Tool" worksheet constitutes the analysis process, which equates to the analysis of codes in more conventional qualitative methodologies.

The information captured in the tools we used was adapted from the UNICEF Human Centered Design (HCD) Toolkit to better reflect disease outbreak contexts. We specifically changed the "Synthesis Tool" worksheet by replacing references to "immunization services" to "overall outbreak response capacities"; reframing "3 AHA! Moments" as "3 most powerful quotes" to capture particularly illustrative or representative reflections rather than only moments of surprise; and removing the section on identifying "2 Things We Must Address" to instead highlight the overall group's perspective on detection, notification, and response. Although the primary author is a trained qualitative researcher, the team deliberately chose this adapted toolkit to build local capacity for qualitative analysis, both by training CDC staff in-country and by providing a clear, simple, and transferable tool for other CDC field offices undertaking 7-1-7 analyses.

## Results

### Assessing timeliness

Detection and notification met 7-1-7 timeliness benchmarks; however, the early response did not meet these benchmarks. Both detection and notification occurred within one day. On January 25, 2023, our records (Table 1) indicate that fishermen in Izabal reported 11 sick pelicans to a regional-level epidemiologist from MAGA, the same day the event was observed. The following day, January 26th, the epidemiologist investigated and notified the national office of MAGA, the entity responsible for action, as well as MSPAS. In contrast to the timeliness of detection and notification, the early response took approximately 34 days. On January 27th, MAGA officials transported two deceased pelicans from Izabal to Guatemala City for HPAI testing. Immediately after transferring samples, active surveillance and biosafety and biosecurity measures were put in place including animal quarantine measures and decontamination. On January 28, a private

**Table 1. Timeline of significant events in the avian influenza outbreak.**

| Timeline | | |
|---|---|---|
| **Response Stage** | **Date** | **Activity** |
| **Detection** | January 25th, 2023 | Regional MAGA epidemiologist receives alert from community of 11 sick pelicans |
| **Notification** | January 26th, 2023 | Regional MAGA epidemiologist investigates outbreak and alerts national MAGA and MSPAS |
| **Early Response** | January 27th, 2023 | MAGA sends officials to bring 2 whole pelicans from Izabal to Guatemala City for testing. |
| | January 28th, 2023 | Private laboratory in Guatemala confirms H5 in pelican sample |
| | January 30th, 2023 | Samples sent to National Veterinary Services Laboratories (NVSL) in Ames, Iowa |
| | February 6th, 2023 | NVSL confirms samples positive for H5N1 |
| | March 1st, 2023 | Approximate date of meeting between MSPAS and Izabal stakeholders on human spillover protocols. |

laboratory in Guatemala confirmed H5 in the samples. Two days later, on January 30, the samples were shipped to the National Veterinary Services Laboratories (NVSL) in Ames, Iowa, which confirmed H5N1 on February 6th. The primary delay stemmed from coordination challenges at the community level. A meeting between MSPAS and stakeholders in Izabal, critical for sharing protocols on recognizing and responding to potential human spillover, did not occur until, at the earliest, March 1, as corroborated by interviews. This delay highlights that while detection and notification were timely, early response benchmarks were not achieved due to the late initiation of essential community engagement.

### Enablers

**Interregional notification and cooperation.** Notifications of avian influenza in neighboring countries from the World Animal Health Information System (WAHIS) placed Guatemalan national authorities on high alert for potential HPAI cases in animals. Launched by the World Organization for Animal Health (WOAH), WAHIS is an early warning and monitoring system that disseminates information about animal health and sends alerts on animal diseases and events in real-time [16]. As one official from MAGA stated,

> "Since November, we were expecting to have the first case. The outbreak was not a big surprise; it was part of surveillance. How we managed the case, it was okay."

The respondent is referencing the multiple HPAI outbreaks in animals across Latin and South America between October 2022 and January 2023 [17]. This knowledge helped the Guatemalan government prepare for a potential outbreak.

**Emergency funding.** For this health event, there was emergency funding available to carry out epidemiological activities. The funds came from the International Regional Organization for Plant and Animal Health (OIRSA), an organization that aims to develop and coordinate programs for the prevention, control, and eradication of diseases and pests in its member countries (Belize, Costa Rica, Dominican Republic, El Salvador, Guatemala, Honduras, Mexico, Nicaragua, and Panama). The emergency funds from OIRSA are $100,000 per year, and if an outbreak needs additional support, the Regional International Council of Ministries of Agriculture (CIRSA) (OIRSA´s highest authority), can allocate an additional $1 million toward a response. For this emergency, MAGA drew on OIRSA to obtain funds for gasoline, vehicles, per diems, meals, and overnight stays for field teams. The availability of OIRSA funds allowed MAGA to respond to health events without needing to submit requests and obtain approvals for other internal funds. These emergency funds, however, have depleted since the conclusion of this health event. Respondents echoed a call for "for the permanent appropriation of emergency funds within their nationally approved budgets.

**Community event-based surveillance (CEBS).** Community event-based surveillance (CEBS) was pivotal to the Guatemalan government's detection of H5N1 in pelicans. CEBS is a public health strategy in which community members participate in detecting unusual health events or behaviors [18]. Typically, this involves trained surveillance informants identifying and reporting events in the community that have public health significance [19]. In Guatemala, community surveillance was developed empirically through past experiences with disease control strategies, such as the eradication of swine fever (2011–2015) and the management of low-pathogenic avian influenza since the 2000s. During these periods, MAGA personnel identified community leaders to establish a passive surveillance network. As a result, communities know whom to contact and where to report disease suspicions or outbreaks, which facilitated early detection in cases. In Izabal, CEBS was enabled by the strong relationship between community members and the regional level MAGA epidemiological team. As one respondent stated,

> "There is a very good communication between these people [fishermen] and the district epidemiologist. Do you know why? We give vaccines and some medicines. They are very grateful, so any action or thing that happens, they call us. We have a few areas where this communication is hard, the areas where we have more indigenous people. [The indigenous people think that] once you call MAGA, they come to kill [their] animals."

The relationship between the community and government was enabled in part by the demographic and linguistic composition of Izabal of Spanish speaking, primarily non-Indigenous populations. In contrast to other parts of the country, with higher percentages of indigenous populations, government institutions are more easily able to build trust and communicate with the local population in Izabal. According to a 2010 report, Izabal is 25.56% indigenous [20]. In the Puerto Barrios municipality of Izabal, where the pelicans were found, only 3% of the population is indigenous [20]. During this outbreak, fishermen from Izabal directly called the regional epidemiologist to inform that person about the 11 dying pelicans. The point of contact for the local fishermen was a MAGA epidemiologist who had established a strong network of contacts after working in the area for 12 years.

**Wide network of laboratories.** The government's ability to collaborate with private laboratories enabled a timely response. The pelican samples were brought from Izabal to Guatemala City on the evening of Friday January 27th. At this time, the national MAGA laboratories were closed for the weekend and the epidemiological team instead took the samples to Laboratorio Sistemas y Equipos, a private laboratory. At the time, there were 4 veterinary diagnostic laboratories that the Guatemalan government had certified and provided official licenses. A polymerase chain reaction (real-time PCR) test revealed that the samples were positive for Avian Influenza H5 on January 28th, 2023, which was the second day of initiating an early response.

### Bottlenecks

**National testing capacity.** Governmental and private laboratories had the reagents to test for H5 but not the N1 subtype in Influenza virus type A [21]. Without the adequate diagnostic equipment to confirm the subtype H5N1 within Guatemala, the state's public health system was reliant on reference laboratories in the United States. The samples (on one Flinders Technology Associates® card with the virus isolated) were shipped on Monday January 30th to NVSL in Ames, Iowa, five days after the initial notification on Wednesday, January 25th. MAGA received notification from NSVL on February 6th 2023 that the samples were positive for H5N1. The lag between shipment and notification was due to severe weather conditions in the USA that impacted the courier service.

**Workforce need.** Workforce needs were noted by all respondents. For example, a respondent from the Department of Surveillance and Control of the Directorate of Epidemiology and Risk Management in MSPAS stated,

> "the main problem is that there are no human resources, it is the same person in charge of surveillance, the same person who goes to the response, the same person to capacity build the team, the same person to be in the field, it's a limitation to our resources in general"

As pointed out, the same national-level epidemiologist who oversees surveillance is also in charge of outbreak responses for all respiratory diseases. The lack of adequate, non-emergency funding to bolster staffing slowed down the responsiveness of MSPAS.

**PPE equipment.** This health event highlighted that the workforce lacked equipment and training for addressing pathogenic and contagious diseases in animals. An interviewee from MAGA's national office was unsure if response teams knew how to don and doff PPE and to safely take and send biological samples. These concerns were echoed at the regional level. One respondent who worked on the ground in Izabal during this health event stated:

> "The truth is that we were not prepared because we did not know what we were facing. The Programa Nacional de Sanidad Avícola (PROSA) response team [that was sent from the national level] had its protective equipment, but it did not have protection for all individuals [involved in the response who lived in the Izabal region]. [Our regional team] didn't have swabs, we didn't even have coolers [to transport the dead pelicans], something so simple...we weren't prepared, the truth is that the team wasn't prepared for what was coming."

After voicing these concerns to colleagues at the national level, this respondent was promised additional supplies but was sent 5 pairs of gloves, a bag of straws, and four overalls to address further surveillance and biosecurity response efforts due to resource constraints. Early in the response, the respondent had to reuse gloves while handling the deceased pelicans because they had already run out of clean gloves.

**Intra-agency and inter-agency communication and planning.** During H5N1 outbreaks, human health sectors should be involved in the response to ensure human spillover can be contained. During this health event, human health sector involvement was the largest barrier to early response completion. One MSPAS official blamed the lack of timely notification from MAGA as the primary reason for the delay in action. As this official recounted,

> "One of the issues we complained about is that the new information did not come to us in a timely way…for this reason the response happened a bit late. We would have wanted to know earlier…we wanted the information when they had the suspicion."

From information obtained from interviews, it appears that MSPAS received this notification between 5 and 7 days after the initial notification to MAGA.

However, our findings indicate that internal communication and planning within MSPAS played even a bigger role in the delayed response. Although MSPAS received the notification, they did not take any on-the-ground action until at least March 1st, when a meeting was held with community members and officials in Izabal. Due to insufficient protocols and preparedness for One Health emergency situations, it took MSPAS several weeks to mobilize the necessary personnel, equipment, and funding. In short, MSPAS was under-resourced and unprepared to respond effectively to the health event. Thus, even with the initial delay in notification from MAGA, there appeared to be significant gaps in internal coordination and limited capacity within MSPAS to initiate a timely, cross-sectoral response aimed at preventing potential disease spillover to humans.

The regional level epidemiological team of MAGA reported that they did not have a clear understanding of their role for both routine surveillance and outbreak response vis-a-vis their counterparts at the national level. Although this did not impact the timeliness of initiating a response, communication issues affected the quality of the response. For this health event, the regional level team worked with the PROSA, a subunit of MAGA that led response efforts. Our respondent at the regional level did not know if PROSA's domain of work included more routine surveillance or if it was limited to responses to health events and outbreaks. When discussing PROSA's response activities, the respondent stated,

> "The next week, another group [from PROSA] came, but no one gave them the information from last week and there was no one who served as their leader. [The week after that], another group [from PROSA] came, but no one gave them the information about the previous week"

This respondent pointed to a disjointed response from PROSA and noted that the area map they were working with was outdated. On it were communities that no longer existed or had moved. Not having consulted the regional epidemiological team of MAGA, the early response may have encountered delays, duplications, and a lack of coverage.

## Discussion

### Timeliness achievements

Our results indicated that detection and notification met the targets of seven and one days, respectively. However, the completion of the early response far exceeded the 7-day benchmark and instead took approximately 34 days (Tables 1 and 2). All steps pertaining to the animal health sector were completed in a timely manner, including deploying investigation/response teams, conducting epidemiological analysis and an initial risk assessment, obtaining laboratory confirmation

**Table 2. 7-1-7 milestone dates.**

| 7-1-7 Milestone dates | Date | Timeliness (in days) | Target (in Days) | Met Target (Yes/No) |
|---|---|---|---|---|
| Emergence | 25 January 2023 | Not Applicable | Not Applicable | Not Applicable |
| Detection | 25 January 2023 | 0 | 7 | Yes |
| Notification | 26 January 2023 | 1 | 1 | Yes |
| Early Response Action Completion | 1 March 2023 | 34 | 7 | No |

of the outbreak etiology, initiating risk communication in affected communities, and establishing a coordination mechanism. The delays in meeting the benchmark were due to issues coordinating with the human health sector (MSPAS), which took 34 days from the date of notification. If there had been human spillover during this outbreak, containment efforts could have been severely hampered as a result.

## Recommendations

This study highlighted two main obstacles in carrying out 7-1-7 assessments. First, bottleneck and enabler analysis could benefit from additional qualitative methods. Previously published work suggests that investigators review documentation of outbreak response efforts and conduct workshops to gather consensus around bottlenecks and enablers [6]. In our study, there was little publicly available documentation on the outbreak response, so we relied almost entirely on qualitatively collected data. We chose to use interviews rather than workshops due to the ethical challenges of carrying out workshops. Workshops would function similarly to focus groups, a qualitative method in which a group of people are asked to discuss and collectively answer questions posed by a moderator. The problem with avian influenza outbreaks, like many other disease outbreaks, is that they are highly politicized health events [22–24]. There are many powerful players placed at risk during an outbreak, such as tourism industries, livestock companies, and politicians. Given the sensitive climate surrounding outbreaks, we decided that focus groups might hinder an honest dialogue between government stakeholders about bottlenecks. This decision was supported by the literature, which notes that focus groups are not effective if participants fear future reprisal or negative impacts because of their honesty [25,26].

Second, this research identified gaps in coordination between the animal and human health sectors, underscoring the need for a stronger One Health approach to outbreak preparedness. Our findings confirm that multinational disease reporting systems (e.g., WAHIS), CEBS, and private-public lab partnerships can enable timely outbreak detection [27–33]. However, as seen in Guatemala, limited cooperation between sectors can slow early response. The Pan American Health Organization and others have emphasized that emerging zoonotic threats are best addressed by strengthening joint surveillance and communication across the human-animal-environmental interface [29].

There are many examples of well-coordinated One Health surveillance approaches in the literature. In Thailand, a One Health surveillance system was coordinated by a single office (the Coordinating Unit for One Health), which had the task of maintaining interagency communication and presiding over the cross-governmental implementation of One Health activities [30]. Taiwan also implements a One Health approach, whereby it integrates wild bird virologic surveillance, domestic avian influenza surveillance, market avian virologic surveillance, swine influenza surveillance, sentinel physician surveillance, human virologic surveillance, and severe case surveillance [31]. These and other models [32] may serve as important starting points for designing a One Health surveillance and outbreak response system.

Efforts to develop a One Health response have been underway in Guatemala. Since 2021, the Centers for Disease Control and Prevention (CDC) have been promoting the One Health approach in Central America, including Guatemala. CDC engaged with the Ministry of Public Health and Social Assistance (MSPAS), the Ministry of Agriculture, Livestock, and Food (MAGA), and the Ministry of Environment and Natural Resources (MARN) to integrate efforts. If the governmental agreement is signed, the One Health Technical Working Group could have an official mandate that facilitates its

operation and cooperation among the ministries. This could enable the establishment of more concrete and effective actions for the detection, notification, and response to zoonotic diseases, thereby improving public health in Guatemala. Institutionalization could also strengthen coordination and communication between the human, animal, and environmental health sectors, promoting a faster and more efficient response to health emergencies.

Finally, Guatemalan outbreak response capacities could benefit from strengthening diagnostic testing capabilities and workforce. During the health event in January 2023, the national MAGA laboratories did not have the capacity to test for H5N1 and relied on the Pan American Health Organization and the Centers for Disease Control and Prevention to test samples in laboratories outside of Guatemala. Following this incident, gaps in testing capacity were identified in the national animal health laboratory capacities, and improvements were made. Strengthening MSPAS's capacity to test for HPAI would improve 7-1-7 metrics and further prepare the agency for outbreaks.

## Limitations

There were several limitations to this study. First, the small sample size may introduce biases by skewing the data to the perceptions of those with whom we were able to interview. Second, it was difficult to access MSPAS and MAGA staff at the local level, who were involved in this outbreak due to high turnover rates. This information from the local level would have provided an even more fine-tuned understanding of the strengths and drawbacks of the outbreak response. Third, the retrospective nature of this study made it difficult to identify exact dates for calculating 7-1-7 metrics. Without any written documentation or paper trails pertaining to the outbreak, interviewee recall issues may have made it difficult to identify the exact dates of early response completion.

Despite these limitations, this paper provides insight into the application of 7-1-7 benchmarks to a zoonotic disease and the Guatemalan government's public health emergency response system. By strengthening One Health approaches to public health, guaranteeing funding mechanisms for public health emergencies, and upgrading laboratory capacities, the Guatemalan government may be able to enhance its ability to provide a timely response to future outbreaks. Future use of the 7-1-7 framework could include a larger pool of respondents, a longitudinal data design, or real-time data collection in order to provide greater nuance into outbreak preparedness in Guatemala and elsewhere.

## Supporting information

**S1 Text. Interview questions.**
(DOCX)

**S1 Table. Rapid qualitative methodology tool.**
(DOCX)

## Acknowledgments

The authors would like to thank all the study participants; the CDC Central America Office for coordinating this project; and the Global Surveillance, Laboratory, and Data Systems Branch, the Global Health Security Team in the CDC's Division for Global Health Protection, and the Influenza Division of the National Center for Immunization and Respiratory Diseases for advising on this project.

## Author contributions

**Conceptualization:** Parsa Bastani, Parminder S. Suchdev.

**Formal analysis:** Parsa Bastani.

**Investigation:** Parsa Bastani, Edgar Bailey Leonardo, Jose Carlos Monzon Fuentes.

**Methodology:** Parsa Bastani.

**Project administration:** Edgar Bailey Leonardo, Jose Carlos Monzon Fuentes, Parminder S. Suchdev.

**Supervision:** Edgar Bailey Leonardo, Cesar Conde Pereira, Emily Zielinski Gutierrez, Parminder S. Suchdev.

**Writing – original draft:** Parsa Bastani.

**Writing – review & editing:** Edgar Bailey Leonardo, Jose Carlos Monzon Fuentes, Cesar Conde Pereira, Emily Zielinski Gutierrez, Parminder S. Suchdev.

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
