## [Decision Letter · Decision Letter 0]

23 Jun 2025

PGPH-D-25-01179

Bottleneck and Enabler Evaluation of Avian Influenza Health Event — Guatemala, January-February 2023

Dear Dr. Suchdev,

Thank you for submitting your manuscript to PLOS Global Public Health. After careful consideration, we feel that it has merit but does not fully meet PLOS Global Public Health’s publication criteria as it currently stands. Therefore, we invite you to submit a revised version of the manuscript that addresses the points raised during the review process.

The reviewers have raised a number of comments and clarifications that should be addressed. There are some comments with suggestions that I think could strengthen the paper: adding a table to make your results easier to follow; providing some more clarity on your discussion on the local context that can help the reader. In addition, please make sure the financial disclosure/conflict of interest and funding statement is correct and standard to the journal's requirements.

We look forward to receiving your revised manuscript.

Kind regards,

Leonor Guariguata, MPH, PhD

Academic Editor

Journal Requirements:

2. We do not publish any copyright or trademark symbols that usually accompany proprietary names, eg (R), (C), or TM (e.g. next to drug or reagent names). Please remove all instances of trademark/copyright symbols throughout the text, including ® on page 15.

3. We note that your Data Availability Statement is currently as follows: [All relevant data are within the paper.]

Additional Editor Comments (if provided):

Reviewers' comments:

Reviewer's Responses to Questions

**Comments to the Author**

1. Does this manuscript meet PLOS Global Public Health’s publication criteria?

Reviewer #1: Partly

Reviewer #2: Partly

2. Has the statistical analysis been performed appropriately and rigorously?

Reviewer #1: N/A

Reviewer #2: N/A

3. Have the authors made all data underlying the findings in their manuscript fully available (please refer to the Data Availability Statement at the start of the manuscript PDF file)?

Reviewer #1: No

Reviewer #2: No

4. Is the manuscript presented in an intelligible fashion and written in standard English?

Reviewer #1: Yes

Reviewer #2: Yes

Reviewer #1: The manuscript is well written and presents a valuable and timely contribution to One Health area. The content is easy to comprehend and the data is relevant. However, the manuscript would benefit from a comprehensive revision, particularly with methodology and data reporting. It would also be great to see further more elaboration on causes/reasons of no 'on-the-ground' action by MSPAS (shared paragraph below). This no-action delayed the effective response even when they managed to adhere to 7-1-7 for detection and investigation initiation phases with limited resource constraint.

"However, internal agency communication and planning in MSPAS appeared to be even more key in delaying the response. Once MSPAS received the notification, they did not take any on-the-ground action until at least March 1st. It took MSPAS several weeks to find the personnel, equipment, and funding. Thus, even with the delay in notification from MAGA, there appeared to be unclear lines of communication within MSPAS to initiate a cross-sectoral response that would protect humans against potential disease spillover."

Reviewer #2: This manuscript describes the application of the 7-1-7 benchmarks to an H5N1 outbreak among pelicans in Guatemala. It appropriately highlights the One Health nature of such an outbreak, given the risk of disease spillover to humans. However, there are several areas that require elaboration before this manuscript is acceptable for publication. A summary of the major areas of improvement are below, with specific comments provided in the attached reviewer manuscript copy:

Introduction - The main issue is to delineate which specific components of the 7-1-7 benchmarks were applied here.

Methods - The authors need to provide more information about how the interviews were conducted (particularly who conducted them and the questions asked). They also need to explain why they chose the UNICEF "Human-Centred Design 4 Health" methodology as opposed to a more traditional qualitative methodology, and they need to provide additional detail about how they used this methodology to organize and synthesize the interview information. Finally, the authors need to justify why the original interview data were not provided with the manuscript.

Results - The authors need to provide high-level interviewee information in a separate table (such as role and agency, as long as the interviewees are not identifiable). The authors also need to provide a table showing how each of the three 7-1-7 steps were met and on what date, broken down by Component. This will help delineate the key rate-limiting steps, such as shipping to NVSL and inter/intra-agency coordination. Also, I'm not clear how the inability to have a meeting between MSPAS and the Izabal stakeholders falls under the poor coordination category.

Discussion - The authors need to further describe how the enablers of outbreak detection and notification compare with the enablers in other countries (and whether those enablers similarly facilitate detection and notification).

Thank you for the opportunity to review.

**Do you want your identity to be public for this peer review?** For information about this choice, including consent withdrawal, please see our Privacy Policy

Reviewer #1: No

Reviewer #2: **Yes: ** Amish Talwar

---

## [Decision Letter · Decision Letter 1]

2 Sep 2025

PGPH-D-25-01179R1

Bottleneck and Enabler Evaluation of Avian Influenza Health Event — Guatemala, January-February 2023

Dear Dr. Suchdev,

Thank you for submitting your manuscript to PLOS Global Public Health. After careful consideration, we feel that it has merit but does not fully meet PLOS Global Public Health’s publication criteria as it currently stands. Therefore, we invite you to submit a revised version of the manuscript that addresses the points raised during the review process.

Please see the few remaining comments on the methodology from Reviewer 1.

We look forward to receiving your revised manuscript.

Kind regards,

Leonor Guariguata, MPH, PhD

Academic Editor

Journal Requirements:

Additional Editor Comments (if provided):

Reviewer #1:

Reviewer #2:

Reviewers' comments:

Reviewer's Responses to Questions

**Comments to the Author**

Reviewer #1: (No Response)

Reviewer #2: All comments have been addressed

publication criteria?

Reviewer #1: Yes

Reviewer #2: Yes

3. Has the statistical analysis been performed appropriately and rigorously?

Reviewer #1: Yes

Reviewer #2: N/A

4. Have the authors made all data underlying the findings in their manuscript fully available (please refer to the Data Availability Statement at the start of the manuscript PDF file)?

Reviewer #1: Yes

Reviewer #2: No

5. Is the manuscript presented in an intelligible fashion and written in standard English?

Reviewer #1: Yes

Reviewer #2: Yes

Reviewer #1: Thanks for all the updates. I have few more requests on methodology section.

Reviewer #2: Thank you for incorporating my suggestions into your manuscript. The paper is considerably improved. I just wanted to note the following minor comments:

-The word "fisherman" in the Introduction and Results section should be plural ("fishermen").

-In the Results section under Assessing Timeliness, you should make it clear upfront that H5 was confirmed in Guatemala City, but the sample needed to go to NVSL to confirm H5N1. I also recommend noting when MSPAS received the notification from MAGA. Also, I think you can delete column 3 in Table 1, which appears redundant.

-In the Discussion section under Recommendations, it's not clear what's the second obstacle in carrying out 7-1-7 assessments - I recommend making this more explicit.

Finally, although sharing interview transcripts with manuscripts is uncommon, I would like to gently push back against the notion that sharing such qualitative research data in general is inappropriate. In fact, it is increasingly expected that interview or focus group transcripts be shared (typically with a data repository - see https://www.pnas.org/doi/10.1073/pnas.2206981120). Choosing not to share such data for fears of interviewee re-identification should be the exception, not the norm, and there are tools available to help mitigate these concerns (see previous article). In the future, I'd encourage the authors to consider whether interview data sharing is a realistic possibility rather than discounting the possibility outright, as appears to have been the case here. For now, I believe it is appropriate to note in the Data Sharing section that raw interview data are unavailable because of privacy concerns.

Thank you for the opportunity to re-review.

**Do you want your identity to be public for this peer review?** For information about this choice, including consent withdrawal, please see our Privacy Policy

Reviewer #1: No

Reviewer #2: **Yes: ** Amish Talwar

---

## [Decision Letter · Decision Letter 2]

27 Oct 2025

Bottleneck and Enabler Evaluation of Avian Influenza Health Event — Guatemala, January-February 2023

PGPH-D-25-01179R2

Dear Professor Suchdev,

We are pleased to inform you that your manuscript 'Bottleneck and Enabler Evaluation of Avian Influenza Health Event — Guatemala, January-February 2023' has been provisionally accepted for publication in PLOS Global Public Health.

Best regards,

Leonor Guariguata, MPH, PhD

Academic Editor

Reviewer Comments (if any, and for reference):

Reviewer's Responses to Questions

**Comments to the Author**

Reviewer #1: All comments have been addressed

publication criteria?

Reviewer #1: Yes

3. Has the statistical analysis been performed appropriately and rigorously?

Reviewer #1: N/A

4. Have the authors made all data underlying the findings in their manuscript fully available (please refer to the Data Availability Statement at the start of the manuscript PDF file)?

Reviewer #1: Yes

5. Is the manuscript presented in an intelligible fashion and written in standard English?

Reviewer #1: Yes

Reviewer #1: Thanks for addressing all the comments. I found it interesting the way you have used UNICEF toolkit framework to analyze the interview transcripts. The description of method section is clearer now but it would have still been nice to clearly visualize the methodological steps followed in this paper which is proposed as 'a unique, simple, and rapid qualitative methodology'.

**Do you want your identity to be public for this peer review?** For information about this choice, including consent withdrawal, please see our Privacy Policy

Reviewer #1: No
